# Molecular Shape-Preserving Au Electrode for Progesterone Detection

**DOI:** 10.3390/s25051620

**Published:** 2025-03-06

**Authors:** Fukuto Soyama, Taisei Motomura, Kenshin Takemura

**Affiliations:** 1Sensing System Research Center, National Institute of Advanced Industrial Science and Technology (AIST), 807-1 Shuku-Machi, Tosu, Saga 841-0052, Japan; soyama.fukuto@aist.go.jp (F.S.); t.motomura@aist.go.jp (T.M.); 2Health Functional Molecular Science Course, Graduate School of Advanced Health Sciences, Saga University, 1 Honjo-Machi, Saga 840-8502, Japan

**Keywords:** Au electrode, progesterone detection, electrochemical biosensor, molecular shape-preserving Au electrode

## Abstract

Quantifying progesterone levels in the body is an important indicator of early pregnancy and health. Molecular shape-preserving electrodes have garnered attention in electrochemical biosensors because they can detect targets without the need for expensive enzymes or antibodies. However, some of the currently used methods typically have low electrode durability. Here, progesterone, for which antibodies are typically expensive, was used to develop a molecular shape-preserving electrode using Au to enhance its long-term stability. The physical properties of the electrodes were characterized using scanning electron microscopy (SEM), the electrochemical surface area (ECSA), and cyclic voltammetry (CV). The specific structure of the electrode demonstrated an electrochemical double layer comparable to that of a smooth Au electrode, confirming its high durability. The detection performance was assessed using CV, square wave voltammetry (SWV), and electrochemical impedance spectroscopy (EIS). The current response to progesterone increased in a concentration-dependent manner, but decreased from the saturated state owing to electrodeposition on the surface. Additionally, electrochemical impedance measurements showed high selectivity compared with hormones with similar structures. The fabricated molecular shape-preserving electrode exhibits an excellent durability, stability, and detection performance, confirming its suitability for long-term use. These findings pave the way to new possibilities for electrode fabrication.

## 1. Introduction

Hormones are an essential part of life and contribute substantially to homeostasis [1,2,3]. Approximately 40 major hormones are secreted by various organs in humans. Seven of these hormones are primarily secreted by the gonads. They are responsible for promoting or inhibiting reproductive growth [4]. Progesterone is a female hormone produced in the corpus luteum of the ovary, which plays an important role in the menstrual cycle and establishing pregnancy [5]. Progesterone deficiency can cause infertility problems, such as luteal insufficiency and polycystic ovary syndrome [6,7]. However, excessive secretion can lead to mental instability, such as premenstrual syndrome [8]. Therefore, quantifying progesterone levels in the body is an important indicator of early pregnancy and health [9].

In addition, monitoring progesterone levels in animals is important for stabilizing reproduction on dairy farms [10]. Progesterone is also an environmental pollutant; therefore, monitoring its levels in ecosystems is important [11]. Routine and versatile progesterone measurements are required. Chromatography and immunoassays are commonly used methods for progesterone detection [12]. Chromatography has high separation power, is suitable for the detection of biological samples, is highly sensitive, and can be used for both quantitative and qualitative analyses. GC-FID (LOD 0.625 ng/mL), which combines gas chromatography with a hydrogen flame ionization detector, is highly specific; however, the sample preparation process is time-consuming [13]. Immunological assays use the antigen–antibody reactions and are highly specific, highly sensitive, and can perform measurements in a short time. The enzyme-linked immunosorbent assay (ELISA) method, which is a type of immunoassay, is highly sensitive (LOD: 8.57 pg/mL), but it has some limits owing to the high cost of enzymes and antibodies [14]. Other methods are expensive in terms of equipment and running costs, and the procedures are complex and require specialized knowledge. In addition, they rely on the quality of the antibodies, and deterioration and storage conditions can affect measurement results [15].

Currently, electrochemical and optical biosensors, which have different advantages compared to the abovementioned measurement methods, are attracting increasing attention [16]. These sensors are easy to use, have short response times, and are highly versatile. Optical biosensors include various methods such as surface plasmon resonance (SPR) spectroscopy, fluorescence resonance energy transfer (FRET), the fluorescence immunoassay (FIA), and resonance light scattering (RLS) [17,18,19]. Each of these methods is a nondestructive analysis; however, they have limitations in terms of cost and sensitivity. Electrochemical biosensors are suitable for detecting biological substances because they are rapid, highly sensitive, cost-effective, and portable [20]. Electrochemical biosensors are generally measured using antibodies and enzymes; for the reasons mentioned above, they are expensive and not widely used [16]. To address this issue, the development of electrodes utilizing molecularly imprinted polymer (MIP) technology has attracted considerable attention [21]. MIPs are created by mimicking and spatially recognizing target substances. The target molecules and functional monomers are copolymerized using a cross-linking agent, and the target molecules are then removed, leaving behind a shape-retaining space. The key advantage of this technology is its ability to detect substances with high sensitivity without relying on antibodies [22]. Molecular imprinting has been used to create functional materials and measure progesterone in urine and serum [22,23]. These methods use measurement techniques requiring redox probes. Reagent-free electrochemical detection has been reported and selective detection has been successfully achieved [24]. However, MIPs are limited in terms of reusability because of the complexity of their synthesis procedure and post-use cleaning. Therefore, there is a need for a new electrode fabrication technique that can be used repeatedly for long periods.

Although molecular shape-preserving electrode fabrication using metals is not currently reported, the fabrication of electrodes via metal deposition offers a promising solution to various challenges. However, conventional deposition methods typically require high temperatures and pressures, rendering them unsuitable for depositing films on biological materials, such as proteins. Magnetron sputtering, which enables low-temperature deposition, is a viable alternative for such applications.

In a previous study, we succeeded in depositing films that mimic biological materials in the order of micrometers. In this study, we focused on a method suitable for detecting progesterone and aimed to replicate its molecular structure with improved precision. A summary of this study is presented in Figure 1. Gold was deposited by sputtering to avoid severe heating of the template by accelerated electron-beam irradiation and the acid degradation of the molecules by plasma irradiation [25]. Electrodeposition of Ni on the Au thin film was used to fabricate electrodes that retained their molecular shape (Figure 1a). The evaluations included measurements of the active surface area, structural analysis (Figure 1b), sensitivity analysis of electrochemical durability (Figure 1c), detection performance (Figure 1d), and target selectivity (Figure 1e). Electrodes that preserve the molecular structure of progesterone capture progesterone without using antibodies or other scavengers. Although its specificity is limited compared to immunoassays, it is effective for qualitative analysis. Our results not only demonstrate new possibilities in biomimetics but also introduce a new approach to electrode fabrication.

## 2. Materials and Methods

### 2.1. Materials and Instrumentation

Progesterone purity ≥ 99%, estradiol, β-estradiol, estrone, potassium chloride, and potassium hexacyanoferrate (III) were purchased from Merck KGaA (Darmstadt, Germany). Sodium hydroxide, 1 M sulfuric acid, ethanol, and acetone were purchased from FUJIFILM Wako Pure Chemical Corporation (Osaka, Japan). The electroforming solution was prepared by mixing 100 mL of 50% *w*/*w* aqueous Ni (II) sulfamate solution (Thermo Fisher Scientific, Waltham, MA, USA).

### 2.2. Apparatus and Measurements

An electrochemical analyzer (ALS 832D, BAS Inc., Tokyo, Japan) was used to evaluate the electrochemical properties, and an AUTOLAB PGSTAT204 (Metrohm AG, Herisau, Switzerland) was used to evaluate the electrochemical sensor performance in terms of SWV and impedance. SEM was performed using a JSM-9100F instrument (JEOL Ltd., Tokyo, Japan) to observe the fabricated electrode surfaces. Surface roughness and height at the nanoscale were investigated using atomic force microscopy (AFM; MFP-3D Origin+, Oxford Instruments, Abingdon-on-Thames, UK).

### 2.3. Au Sputtering on Progesterone

#### 2.3.1. Progesterone Au Electrode (P4AuEL) Fabrication

Figure 2 shows the procedure used to prepare the P4AuELs. First, a 10 mM progesterone solution (ethanol solvent) was dropped onto a smooth glass substrate and allowed to dry (Figure 2a). A thin Au film with attached progesterone molecules was then deposited on the glass substrate using sputtering (Figure 2b). Once the glass samples were introduced into the chamber, the moisture content within them increased the base pressure to 2 × 10^−2^ Pa. The argon gas pressure and flow rate were maintained at 0.13 Pa and 10.0 sccm, respectively. The Au target used in the deposition process had a purity of 99.99%, diameter of 50 mm, and thickness of 2.5 mm. The distance between the target and substrate surface remained fixed at 50 mm, and the deposition time was 10 min. A thin film, which mimicked the molecular structure of progesterone using Au, was converted into a thick film using Ni to form an electrode. First, the thin film was vacuum dried. Next, this thin film was electrodeposited on the thin film at −1.9 V for 3 h in an Ni solution at approximately 75 °C and chrono amperometry (CA), using this thin film as the working electrode, an Ni plate as the counter electrode, and Ag/AgCl as the reference electrode (Figure 2c). The thickened Au-Ni composite was stripped from the glass substrate (Figure 2d) and washed with acetone and ethanol to remove progesterone from the Au (Figure 2e).

#### 2.3.2. Electrochemical Progesterone Detection

For progesterone, a 10 mM solution was prepared with ethanol and diluted to 10 μM with 10 mM PBS. A 2-fold dilution was then made with 10 mM PBS to prepare a 5 μM solution. A 5-fold dilution was then made to prepare a 1 μM solution. Dilutions were repeated until a final concentration of 1 pM was obtained. Ethanol was added to all the prepared solutions to achieve a final ethanol concentration of 0.1 vol%. For square wave voltammetry (SWV) measurements, progesterone concentrations were measured in decreasing order from 0 M. Prior to impedance measurements, cyclic voltammetry (CV) was performed for four cycles in the voltage range of −0.3 to0.5 V. After measurement, the electrochemical cell and electrode surfaces were cleaned with ethanol. Impedance measurements were performed in the frequency range of 100,000 Hz–0.1 Hz. All measurements were taken three times. For the analysis of the measured data, the “Fit and Simulation” function of the Nova analysis tool was used to apply a circuit that considers the electrochemical double layer. Other hormones with molecular structures similar to that of progesterone were measured using similar procedures.

## 3. Results and Discussion

### 3.1. Structural Properties of P4Au

Figure 3 shows the SEM and AFM results before and after thickening of the Ni layer after sputtering Au on a glass substrate. Figure 3a shows an SEM image after Au was sputtered onto the glass substrate. The Au thin film formed a nanoparticle structure, which was assembled in a complex manner by subsequent Ni electrodeposition to form a durable electrode structure. A similar trend was observed in the scaled-up SEM image, which confirmed the macroscopic Au and Ni composite (Appendix A). Figure 3b shows an SEM image of the Au surface when the glass substrate was separated and removed after Ni thick-film deposition. The Ni side exhibits a nanoparticle structure, whereas the Au side, which adheres to the glass substrate, maintains a smooth surface. This smooth surface may contain regions that partially mimic the molecular structure of progesterone. A macroscopically smooth surface was also observed in the scaled-up SEM image (Appendix A). Figure 3c,d show the AFM images of the Au surface after Ni thickening. The Au surface can be observed to be sufficiently smooth, with irregularities of ±5 nm. This result is consistent with the SEM image in Figure 3b. Atomic–scale sputter deposition has been reliably demonstrated [26,27,28]. In addition, image analysis of the AFM data confirmed the presence of nanosized or smaller microscopic structures, suggesting that these structures may selectively adsorb progesterone. In contrast, when we reviewed AFM results with an enlarged scale, shown in Appendix A, an unevenness of approximately ±20 nm was observed, but this was also within the range where the surface could be sufficiently smooth.

Appendix A shows the results of the image analysis based on the AFM data. The 3D view in Appendix A matches the unevenness shown in Figure 3d, confirming the reliability of the image analysis results. Appendix A presents a two-dimensional (2D) version of the 3D view. Several microscopic areas were observed, as indicated by the dark blue color. Furthermore, Appendix A shows a profile with the scale enlarged, and the surface irregularities are found to be approximately 0.1–0.3 nm, which is nearly the same size as the interatomic distance. The attachment of progesterone molecules to these bulks is thought to increase the resistance of the electrode. However, in addition to the microscopic bulks, there were also larger bulks, which suggested areas where the progesterone molecules overlapped. Deposition on glass substrates was demonstrated to preserve the shape after Ni electroforming [29,30]. These results suggest that there are partial sites on the Au surface where molecules can be adsorbed and that this property could be used to quantitatively and selectively detect progesterone.

### 3.2. Electrochemical Properties of P4AuEL

Figure 4 shows the electrochemical properties of P4AuEL. Figure 4a presents the CV in 10 mM KCl containing 50 mM potassium ferricyanide as a redox probe to examine the electrode response. The electrodes fabricated in this study were compared with a glassy carbon electrode (GCEL, BAS Inc., Tokyo, Japan), known for its electrochemical inertness and physical and chemical stability, as well as a smooth gold electrode (AuEL), which has a similar composition to P4AuEL [31,32]. For P4AuEL, the redox peaks attributed to potassium ferricyanide and potassium ferrocyanide were observed at higher potentials than those for AuEL and GCEL. This may have been due to the nanostructured electrode surface, which resulted in extensive peaks. These characteristics suggest that P4AuEL performs sufficiently well as a sensing electrode. In addition, the surface roughness of Au electrodes, which are imperfectly smooth films formed by sputtering, increases with successive CV cycles [32]. In contrast, P4AuEL exhibited no differences in peaks associated with potassium ferricyanide during continuous cycling, confirming its high stability and reproducibility as an electrode. The active surface area of P4AuEL was estimated using the electrochemical surface area (ECSA) method by calculating its capacitance. Figure 4b illustrates CV conducted in a 10 mM PBS solution within a voltage range of 0–0.1 V and a sweep rate varying from 10 to 100 mV/s. The results of the AuEL ECSA are shown in Appendix A. Capacitance C_dl_ was determined by linearly plotting each current density against the sweep speed at ΔJ = 0.05 V (Figure 4c). P4AuEL has a capacitance comparable to that of a smooth-surface AuEL. Generally, the larger the surface roughness of an electrode, the larger its electrochemical surface area [33]. However, P4AuEL exhibited the opposite behavior. The current density of P4AuEL was suppressed despite the large specific surface area of the electrode. This is also related to the peak current values in Figure 4a, indicating that P4AuELs have better current stability than AuELs. To confirm the durability of the electrode, 100 CV cycles were conducted in 10 mM PBS, spanning a voltage range of −0.3 to 0.5 V at a sweep rate of 50 mV/s (Figure 4d). Within the measured potential range, no significant change was observed in the number of cycles, confirming the stability of the electrode. A similar trend was observed for AuELs, further validating the durability and effective active surface area of P4AuEL. Figure 4e shows the impedance measurements of P4AuEL and AuEL in 10 mM PBS (0.1% EtOH); the resistance of P4AuEL was slightly higher than that of AuEL. This result is related to the high potential shift of the redox potential peak observed in Figure 4a, which may be due to the nanostructuring of the surface. Furthermore, this result is consistent with the data presented in Figure 4b,c, which indicate that the active surface areas of P4AuEL and AuEL are almost the same. Accordingly, it can be concluded that the electrochemical properties of P4AuEL are generally similar to those of AuEL, and that the difference in the redox potential is mainly due to the difference in the surface topography. Figure 4f shows the results of the SWV measurements of P4AuEL. The measurement conditions were performed with a voltage range of 0 to −0.75 V, Pt as a counter electrode, and PBS as a buffer. Progesterone concentrations were measured at every 10 fold dilution from 1 pM to 1 µM. The results showed a concentration-dependent enhancement of the peak at 0.3 V, suggesting that the characteristic peak intensity increased as progesterone interacted with the Au surface. Progesterone also acted on the Au surface, saturating at 10 nM. In the present measurements under no-washing conditions, 10 nM was considered the concentration at which the interaction of progesterone with the P4AuEL surface was maximal.

### 3.3. Electrochemical Selectivity of P4AuEL for Progesterone

To evaluate progesterone quantification using P4AuEL, electrochemical impedance spectroscopy (EIS) was conducted for progesterone concentrations ranging between 0 M and 10 μM (Figure 5). Before each measurement, four CV cycles were performed in buffer solution, with the results of the second cycle presented in Figure 5a. A gradual decrease in the characteristic peak current of the gold electrode was observed as the progesterone concentration increased. This trend is consistent with previously reported results from the literature on MIP [22,34,35]. It is suggested that the electrode surface plays a similar role to that of an antibody and that charge transfer is inhibited by the adsorption of progesterone. This decrease was attributed to increased resistance caused by progesterone adsorption on the electrode surface. The characteristic Au peak current values for each concentration were plotted (Appendix A), revealing a non-linear decrease in current, which in turn suggests that progesterone adsorption on the electrode surface follows a logarithmic trend. Impedance measurements for P4AuEL and AuEL under identical conditions are shown in Figure 5b,d. For AuEL, no significant change in resistance was observed with varying progesterone concentrations, likely due to its smooth surface, which lacks specific adsorption sites for progesterone. In contrast, P4AuEL exhibited a concentration-dependent increase in resistance (Figure 5c), with a correlation coefficient exceeding 0.997 indicating a linear relationship. A similar trend was reported for quantitative analysis by EIS using antibody-modified electrodes [36]. Quantitative detection by EIS using MIP without the use of antibodies has also been confirmed [37]. Based on these reports, the linear concentration dependence of P4AuEL suggests that progesterone is adsorbed on the electrode surface. This increase is ascribed to electrodeposition occurring on the electrode surface. While this observation partially corroborates the findings in Figure 5a, it contrasts with the non-linear decrease in peak current, suggesting that the progesterone-free solution used during CV measurements may have left residual progesterone on the electrode surface in a non-linear manner as the concentration increased. The nanoscale spatial structure on the P4AuEL surface likely facilitated reliable progesterone adsorption detection. Additionally, impedance measurements were conducted for hormones with chemical structures similar to progesterone, and the results, along with the corresponding resistance graphs, are presented in Figure 5e,f. For these hormones, resistance changes were minimal compared to the control. This indicates that while these substances interact with the P4AuEL surface, only progesterone caused a significant resistance increase. These findings highlight the strong and selective interaction between progesterone and the P4AuEL surface. These results suggest that by preserving the molecular structure of progesterone at the electrode interface, a certain degree of quantitative and qualitative functionality can be exhibited in solutions containing few foreign substances. However, the signal difference with estradiol is small and does not show as much selectivity as the immunoassay using antibodies for similarity detection targets. This indicates that it is difficult to use metal imprinting electrodes in solutions containing a large number of structurally similar materials, such as human serum, which is an actual specimen.

The performance of this sensor was compared with those of other progesterone sensors to verify the advantages of this study (Table 1). The EIS showed similar sensitivity, indicating that the progesterone-mimicking electrode surface is comparable to other sensing surfaces including MIP. This suggests that P4AuEL could offer an alternative to other sensing approaches.

## 4. Conclusions

In this study, we used magnetron sputtering to fabricate composite Au and Ni electrodes that mimic molecular structures. Particular emphasis was placed on the fabrication of electrodes suitable for biomaterials, and their electrochemical performances were evaluated. The surface analysis revealed that nanoscale deposition was achieved, partially revealing a bulk structure of 0.1–0.3 nm. However, several challenges remain for the analysis of surfaces at the atomic level. The notable features of this study include the possibility of label-free detection, selective detection of the target substance without the use of polymeric materials, and the novelty of the deposition technique at the angstrom level. The fabricated electrode suppressed the current density while maintaining an active surface area equivalent to that of a Au electrode. This can be attributed to the nanostructuring of the electrode surface. In addition, impedance measurements revealed a concentration-dependent increase in the resistance of the fabricated electrodes. Only the target substance was electrodeposited, showing a higher selectivity than that of the smooth Au electrode, which showed no change. This result suggests that the electrode has a specific surface structure. However, compared to commercially available immunoassays for female hormones, the performance was not sufficient to be adapted to spike samples, especially in the area of selectivity. This was due to the lack of structural preservation performance of gold, the deposition material, for the Ålevel target of the molecular structure. To utilize this electrode, it is necessary to remove many foreign substances from the solution containing the detection target, as many as possible. An important result of this study is the improved stability of the electrode compared to that of polymer materials, which makes it suitable for long-term use. Another important feature is the potential for widespread use because of the low cost when avoiding using antibody materials. Furthermore, the system is expected to contribute to green chemistry as no highly toxic solvents or environmental pollutants are used in the deposition and detection processes.

## Figures and Tables

**Figure 1 sensors-25-01620-f001:**
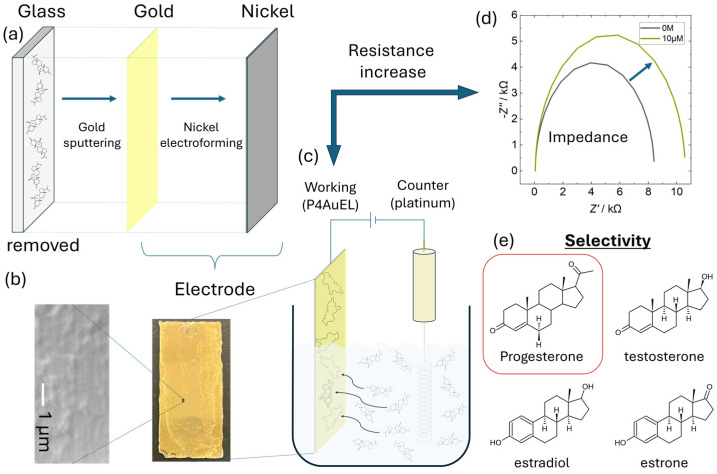
Schematic diagram of this study. (**a**) Electrode configuration. A thin Au film is formed on a glass substrate by sputtering. Ni is then electrodeposited on the Au film, and the glass substrate is removed. (**b**) SEM image of the fabricated electrode surface. (**c**) Schematic of the electrochemical measurement system. (**d**) Impedance measurement of progesterone. The arrows indicate that the impedance measurement is measuring the electrical resistance of the working pole interface. Increased resistance was observed. (**e**) Other selectivity was shown by comparison with progesterone-like structures.

**Figure 2 sensors-25-01620-f002:**
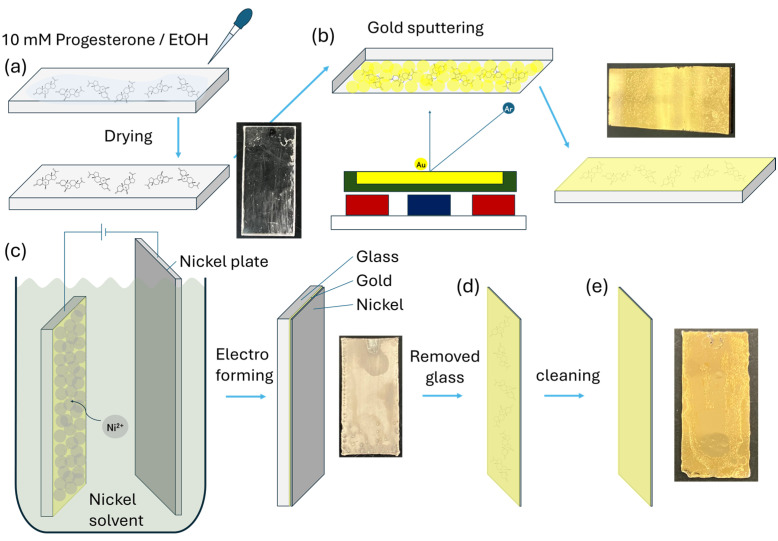
Schematic of the electrode fabrication. (**a**) We drop dried 10 mM progesterone–ethanol on a glass slide substrate. (**b**) Sputtering system (Au deposition). (**c**) Ni electrodeposition on the Au glass slides. (**d**) The glass substrate was removed. (**e**) The progesterone remaining on the Au film was washed with an organic solvent.

**Figure 3 sensors-25-01620-f003:**
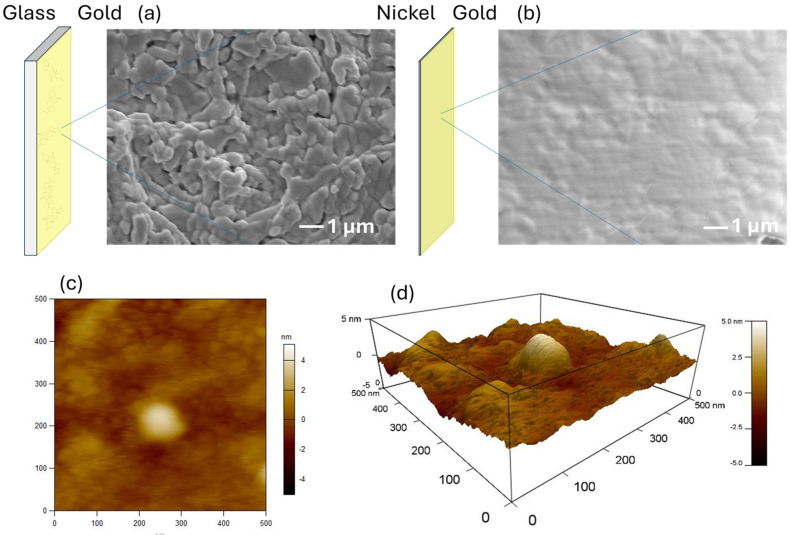
Analysis of the electrode surface. (**a**) SEM image before Ni electrodeposition. (**b**) SEM image after Ni electrodeposition. (**c**) AFM results for a 500 nm square after electrodeposition with Ni. (**d**) Three-dimensional view of a 500 nm square AFM after Ni electrodeposition.

**Figure 4 sensors-25-01620-f004:**
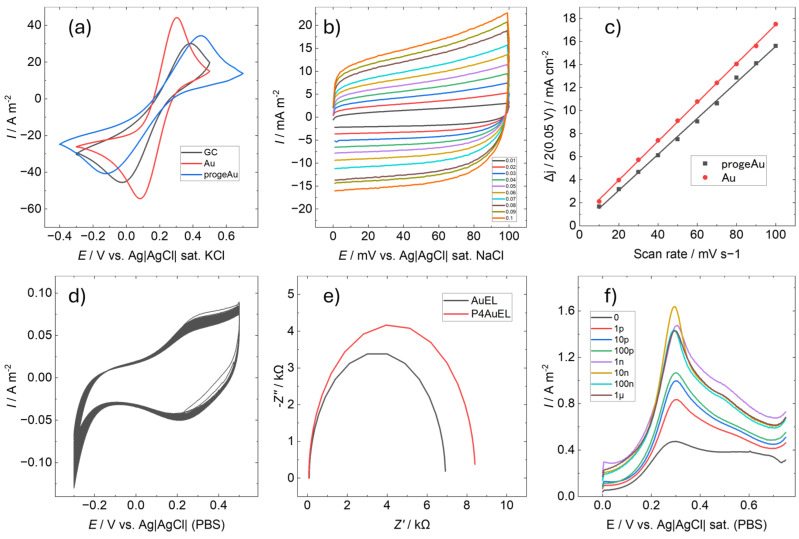
Electrochemical characterization. (**a**) Cyclic voltammetry (CV) of GC, AuEL and P4AuEL in 10 mM KCl containing 50 mM potassium ferricyanide. (**b**) CV of P4AuEL with a varying scan rate. (**c**) Electrochemically active surface area. (**d**) CV of 100 cycles of P4AuEL. (**e**) Impedance measurements of P4AuEL and AuEL. (**f**) Square wave voltammetry from 0 to 1 μM progesterone in P4AuEL.

**Figure 5 sensors-25-01620-f005:**
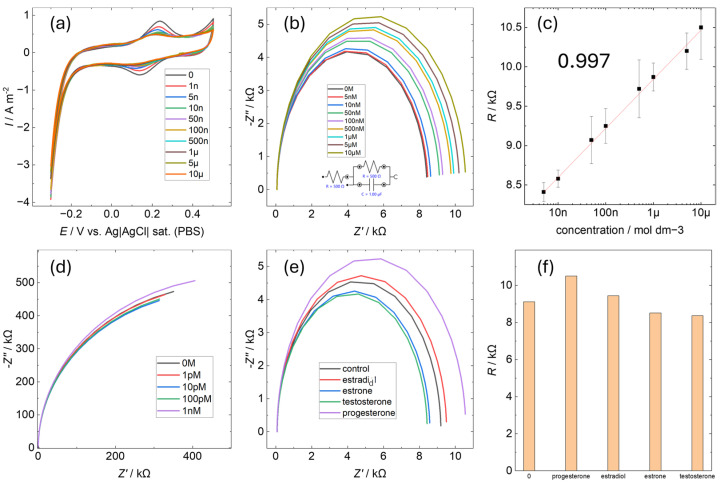
Electrochemical detection performance evaluation. (**a**) Cyclic voltammetry between 0 and 10 μM progesterone in P4AuEL before impedance. (**b**) Impedance measurements between 0 and 10 μM progesterone in P4AuEL. (**c**) Plot of the resistance values corresponding to the concentration changes in (**b**). (**d**) Impedance measurements from 0 to 1 nM progesterone in AuEL. (**e**) Impedance measurements of progesterone-like structures (estradiol, estrone, and testosterone) in P4AuEL. (**f**) Comparison of the respective resistance values in (**e**).

**Table 1 sensors-25-01620-t001:** Performance comparison of progesterone biosensor.

Method	Functional	Sample	LOD	Range	Ref.
HPLC	MIP	Human urine	0.47 ng/mL	0.47–1.15 ng/mL	[38]
EIS	Aptamer	Buffer	0.9 ng/mL	10–60 ng/mL	[39]
SPR	NIR QD	Buffer	5 nM	5 nM–126 mM	[40]
SWV	GCEL	Buffer	68 nM	0.22–14 μM	[41]
EIS	P4AuEL	Buffer	1.73 nM	4.05 nM–34.2 μM	*

*: this work.

## Data Availability

The datasets generated and/or analyzed during the current study are available from the corresponding author on reasonable request.

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
