# Peer review of "Molecular Shape-Preserving Au Electrode for Progesterone Detection"

_sensors, 2025, doi:10.3390/s25051620_

Round 1
Reviewer 1 Report
Comments and Suggestions for Authors
This manuscript (sensors-3475847) presents a molecular shape-preserving electrode for the detection of progesterone. The issues should be further considered are as follows:
1. A brief description of the current development of molecular shape-preserving electrodes needs to be added in the Introduction section.
2. In Fig. 2b, the temperature should be specified during the sputtering of Au film on the glass substrate, as a high sputtering temperature is unsuitable for depositing films on proteins.
3. How to ensure that the progesterone molecules do not overlap with each other on the glass substrate? Otherwise, the molecular shape formed on the Au film does not match the actual progesterone shape.
4. It is suggested to mark possible progesterone molecule shapes in the AFM image to verify the successful preparation of the sensor structure.
5. It is necessary to optimize the preparation process and parameters of the sensor, such as the concentration of progesterone dropped onto the glass substrate.
6. The sensing performance of the sensor needs to be compared with other progesterone sensors in a table format to verify the advantages of this work.
7. The progesterone detection results of AuEL should be added for comparison to prove that the signal changes are caused by the molecular shape.
8. More sensing performance results need to be presented, such as the stability, durability and suitability for long-term use which are mentioned in Abstract. More details of sensing mechanism should be analyzed and discussed.
Author Response
Thank you for your valuable peer review. We have carefully addressed the issues you raised in the manuscript. All changes have been annotated with comments in blue and highlighted in green for direct corrections. We appreciate your insights, which have undoubtedly improved the quality of our work.
This manuscript (sensors-3475847) presents a molecular shape-preserving electrode for the detection of progesterone. The issues should be further considered are as follows:
- A brief description of the current development of molecular shape-preserving electrodes needs to be added in the Introduction section.
Reply: Thank you for pointing this out. We have added the following text based on your comments.
Molecular shape-preserving electrode fabrication using metals is not currently reported.
- In Fig. 2b, the temperature should be specified during the sputtering of Au film on the glass substrate, as a high sputtering temperature is unsuitable for depositing films on proteins.
Reply: As you have indicated, it is possible that progesterone molecules overlap with the glass substrate. We have not yet reached this analysis, but we have clearly marked the areas where the progesterone molecules adhered, which we have added to Fig. S2, as we have answered in question '4.’ We revised in MS as follows:
The attachment of progesterone molecules to these bulks is thought to increase the resistance of the electrode. However, in addition to the microscopic bulks, there were also larger bulks, which suggested areas where progesterone molecules overlapped.
Also, the sample temperature during sputtering has been proven in a previously published paper. The substrate surface temperature has been proven to be less than 40 degrees Celsius.
Metal deposition and shape reproduction at biological temperatures on cell-level samples | Scientific Reports
- How to ensure that the progesterone molecules do not overlap with each other on the glass substrate? Otherwise, the molecular shape formed on the Au film does not match the actual progesterone shape.
Reply: The most effective method is to create a monolayer, such as a self-molecular membrane, via chemical bonding. However, when detecting progesterone, which has complex dynamics such as polycation in solution, it would be better to adsorb progesterone in such a way that it can bind in a random direction. Therefore, in this study, a detection substrate was prepared using simple adsorption.
- It is suggested to mark possible progesterone molecule shapes in the AFM image to verify the successful preparation of the sensor structure.
Reply: To address the comments received, image analysis of the AFM image was performed. Progesterone itself is Å level in size, and it is difficult to analyze a valid AFM value for thickness. However, we extracted several characteristic shapes from the image analysis. We have added the analysis to Supporting Figure 2. Please check it.
- It is necessary to optimize the preparation process and parameters of the sensor, such as the concentration of progesterone dropped onto the glass substrate.
Reply: The size of the glass substrate was 12 mm × 25 mm and the concentration and amount of progesterone saturating the glass substrate uniformly was calculated from the molecular size of progesterone. As a result, the concentration of progesterone dropped onto the glass substrate was 10 mM progesterone/EtOH.
- The sensing performance of the sensor needs to be compared with other progesterone sensors in a table format to verify the advantages of this work.
Reply: Thank you for pointing this out to us. We have added a table1 to the manuscript by comparing it with other papers. Please check it.
Table1. Performance comparison of progesterone biosensor.
Method |
Functional |
Sample |
LOD |
Range |
Ref. |
HPLC |
MIP |
Human urine |
0.47ng /mL |
0.47-1.15 ng / mL |
38 |
EIS |
Aptamer |
Buffer |
0.9 ng/mL |
10-60ng/mL |
39 |
SPR |
NIR QD |
Buffer |
5nM |
5 nM-126 mM |
40 |
SWV |
GCEL |
Buffer |
68nM |
0.22-14μM |
41 |
EIS |
P4AuEL |
Buffer |
1.73 nM |
4.05nM-34.2 μM |
* |
* this work.
- The progesterone detection results of AuEL should be added for comparison to prove that the signal changes are caused by the molecular shape.
Reply: The impedance measurements for P4AuEL and AuEL under identical conditions are shown in Fig. 5b and 5d, respectively. For AuEL, no significant change in resistance was observed with varying progesterone concentrations, likely because of its smooth surface, which lacks specific adsorption sites for progesterone. - More sensing performance results need to be presented, such as the stability, durability and suitability for long-term use which are mentioned in Abstract. More details of sensing mechanism should be analyzed and discussed.
Reply: To confirm the durability of the electrode, 100 CV cycles were conducted in 10 mM PBS, spanning a voltage range of -0.3 to 0.5 V at a sweep rate of 50 mV/s (Fig. 4d). Within the measured potential range, no significant change was observed in the number of cycles, confirming the stability of the electrode. A similar trend was observed for the AuELs, further validating the durability and effective active surface area of the P4AuEL.
Reviewer 2 Report
Comments and Suggestions for Authors
The development of highly sensitive sensors suitable for multiple use is a very urgent task. But I have a number of questions and comments to the authors:
When registering impedance hodographs, it is necessary to indicate which equivalent circuit was used?
How many parallel measurements were carried out?
It would be good to check the correctness of the measurements using the "entered-found" method.
Author Response
Thank you for your valuable peer review. We have carefully addressed the issues you raised in the manuscript. All changes have been annotated with comments in blue. We appreciate your insights, which have undoubtedly improved the quality of our work.
The development of highly sensitive sensors suitable for multiple use is a very urgent task. But I have a number of questions and comments to the authors:
When registering impedance hodographs, it is necessary to indicate which equivalent circuit was used?
Reply: We have added a schematic of the drawing you have mentioned. Please refer to the revised manuscript.
How many parallel measurements were carried out?
Reply: In this case, all measurements were taken at n=3. The electrodes were three metal-imprinted electrodes fabricated using the same procedure.
It would be good to check the correctness of the measurements using the "entered-found" method.
Reply: Progesterone was the substance measured in this case, but it was possible to confirm this by using a commercially available test kit. However, it was difficult to obtain the kit within the peer-review response period, and we were unable to prepare the data. We confirmed the selectivity of the fabricated sensor using other hormones. Although there is still room for improvement in quantitation, the low-temperature deposition technique demonstrates the feasibility of a biosensor that does not require antibodies.
Reviewer 3 Report
Comments and Suggestions for Authors
The authors have presented an interesting idea with clinical interest. The electrochemical sensors have found many valuable applications in different analytical purposes in pharmaceutical analysis. The manuscript has been well prepared, based on a recent literature background. The article has a sound scientific basis and sheds new light on progesterone detection, although some more or less important aspects should be improved by the authors:
- The fabrication procedure is well described but the applicability of the electrode as working in the analytical field should be improved. Based on the article title I expected a more analytical approach with some aspects of the validation procedure such as accuracy, linearity range, limits etc.
- The authors should clarify why they chose SWV voltammetry with selected parameters, giving no satisfactory reason for not applying i.e. DP voltammetry. The current vs. potential plots are missing.
- Can EIS should be used as a method for progesterone detection? EIS is designed for sensor characterization. Given the course of the calibration curves (Fig. 5c), this is highly questionable, especially at the extremes.
- I could not find the information on the purity of reagents. Progesterone is a naturally occurring chiral steroid and is known to be polymorphic, as described in the literature. This fact should be noted and checked is/or how it affects the detection of progesterone and the sensor fabrication phase (also specificity).
In my opinion, the manuscript is unsuitable for publication in its current form.
Author Response
The authors have presented an interesting idea with clinical interest. The electrochemical sensors have found many valuable applications in different analytical purposes in pharmaceutical analysis. The manuscript has been well prepared, based on a recent literature background. The article has a sound scientific basis and sheds new light on progesterone detection, although some more or less important aspects should be improved by the authors:
Thank you for your valuable peer review. We have carefully addressed the issues you raised in the manuscript. All changes have been annotated with comments in blue and highlighted in yellow for direct corrections. We appreciate your insights, which have undoubtedly improved the quality of our work.
- The fabrication procedure is well described but the applicability of the electrode as working in the analytical field should be improved. Based on the article title I expected a more analytical approach with some aspects of the validation procedure such as accuracy, linearity range, limits etc.
Reply: In a previously published article, SWV was evaluated as having the performance of a non-enzymatic electrochemical sensor in the detection of progesterone4. - The authors should clarify why they chose SWV voltammetry with selected parameters, giving no satisfactory reason for not applying i.e. DP voltammetry. The current vs. potential plots are missing.
Reply: Table 1 was created and referred to with regard to detection performance. The correlation coefficients are also shown in Fig. 5(c). - Can EIS should be used as a method for progesterone detection? EIS is designed for sensor characterization. Given the course of the calibration curves (Fig. 5c), this is highly questionable, especially at the extremes.
Reply: There have already been reports on bio-sample detection using EIS. For materials with strong insulating properties, such as proteins, the EIS signal change is more reliable and concentration dependent. Progesterone was the target in this study, not the viral particles, as in the references, resulting in a less stable rate of signal change. This is mainly due to the randomness of the immobilization on the electrode. If the chemical structure is highly polar, the application of an electric field is effective in attracting the compound to the electrode; however, this is not the case here.
Electrical pulse-induced electrochemical biosensor for hepatitis E virus detection | Nature Communications
- I could not find the information on the purity of reagents. Progesterone is a naturally occurring chiral steroid and is known to be polymorphic, as described in the literature. This fact should be noted and checked is/or how it affects the detection of progesterone and the sensor fabrication phase (also specificity).
Reply: Purity ≥99%. It is a chiral steroid hormone, but is not taken into account when measuring detection, as the same reagents are used. We added purity information of progesteron in materials section.
Round 2
Reviewer 1 Report
Comments and Suggestions for Authors
Accept it as it is.
Author Response
Thank you for your reviewing.
Reviewer 3 Report
Comments and Suggestions for Authors
Dear Authors,
Thank you for your replies. Some of my concerns have been addressed, but some have been overlooked or misunderstood. First of all, I appreciated your design and the work that has been done in fabricating the new sensor. However, It seems that the article was not to be thoughtful. There is a lack of correspondent title, the aim of the work is not highlighted and the results should be better described. As I previously wrote the application of the proposed sensor is not fully studied. There is a problem with precision, linearity range, and sensitivity, so the sensor could not work properly in the detection process. I suggest focusing on the sensor itself and showing its applicability in only limited qualitative analysis or characteristic qualitative analysis including its uncertainty level. This fact should also mentioned in the introduction section, included in the results and highlighted in the conclusions section. You pointed out a similar paper on HEV detection although in this work the detection is more accurate and in this case, qualitative analysis is more reliable. Moreover, progesterone monitoring requires preferably quantitative analysis i.e. in the field of environmental or clinical analysis.
Fig. 5e should be estradiol instead of estragiol
Author Response
Thank you for your replies. Some of my concerns have been addressed, but some have been overlooked or misunderstood. First of all, I appreciated your design and the work that has been done in fabricating the new sensor. However, It seems that the article was not to be thoughtful. There is a lack of correspondent title, the aim of the work is not highlighted and the results should be better described. As I previously wrote the application of the proposed sensor is not fully studied. There is a problem with precision, linearity range, and sensitivity, so the sensor could not work properly in the detection process. I suggest focusing on the sensor itself and showing its applicability in only limited qualitative analysis or characteristic qualitative analysis including its uncertainty level. This fact should also mentioned in the introduction section, included in the results and highlighted in the conclusions section. You pointed out a similar paper on HEV detection although in this work the detection is more accurate and in this case, qualitative analysis is more reliable. Moreover, progesterone monitoring requires preferably quantitative analysis i.e. in the field of environmental or clinical analysis.
Reply: Thank you for pointing this out. Indeed, the data presented in this study are limited, and many points need to be clarified in the future. Therefore, we have reiterated the missing functionality of the sensor in the introduction, results, and conclusions. The revised MS is highlighted in green.
PAGE 3、LINE 97-99
Electrodes that preserve the molecular structure of progesterone capture progesterone without using antibodies or other scavengers. Although its specificity is limited compared to immunoassays, it is effective for qualitative analysis.
PAGE 8、LINE 327-334
These results suggest that by preserving the molecular structure of progesterone at the electrode interface, a certain degree of quantitative and qualitative functionality can be exhibited in solutions containing few foreign substances. However, the signal difference with estradiol is small and does not show as much selectivity as the immunoassay using antibodies for similarity detection targets. This indicates that it is difficult to use metal imprinting electrodes in solutions containing a large number of structurally similar materials, such as human serum, which is an actual specimen.
PAGE 10、LINE 363-368
However, compared to commercially available immunoassays for female hormones, the performance was not sufficient to be adapted to spike samples, especially in the area of selectivity. This was due to the lack of structural preservation performance of gold, the deposition material, for the Å-level target of the molecular structure. To utilize this electrode, it is necessary to remove many foreign substances from the solution containing the detection target as much as possible.
Fig. 5e should be estradiol instead of estragiol
Reply: We revised the drawing as indicated. Please check them.